# Glycosylated SARS-CoV-2 RBD Antigens Expressed in Glycoengineered Yeast Induce Strong Immune Responses Through High Antigen–Alum Adsorption

**DOI:** 10.3390/biom15081172

**Published:** 2025-08-15

**Authors:** Ai Li, Tiantian Wang, Bin Zhang, Xuchen Hou, Peng Sun, Hao Wang, Huifang Xu, Min Tan, Xin Gong, Jun Wu, Bo Liu

**Affiliations:** 1Department of Microorganism Engineering, Beijing Institute of Biotechnology, Beijing 100071, China; liai980909@163.com (A.L.); wangtiantian@bmi.ac.cn (T.W.); zb008513@126.com (B.Z.); hoxuch@163.com (X.H.); sp20@mails.tsinghua.edu.cn (P.S.); king13298118546@163.com (H.W.); xuhuifang9169@163.com (H.X.); tanmin980707@126.com (M.T.); 13693331905@163.com (X.G.); 2Institute of Physical Science and Information Technology, Anhui University, Hefei 230601, China; 3Department of Biopharmaceutical Science, Shanghai Ocean University, Shanghai 201306, China; 4Medical College, Hubei Enshi College, Enshi 445000, China

**Keywords:** phosphomannose, aluminum hydroxide adjuvant, glycoform modification, COVID-19 vaccines

## Abstract

Glycosylation plays a pivotal role in regulating the functions and immunogenicity of antigens. Targeting the receptor-binding domain (RBD) of the spike protein (S protein) of SARS-CoV-2, we examined the impact of different glycoforms on RBD antigen immunogenicity and the underlying mechanisms. IgG-specific antibody titers and pseudovirus neutralization were compared in mice immunized with RBD antigens bearing different glycoforms, which were prepared using glycoengineering-capable *Pichia pastoris* and mammalian cell expression systems with distinct glycosylation pathways. The glycosylation impacted the surface charges of the RBD antigen, and influenced its adsorption onto alum. This may further lead to variations in the antigen’s immunogenicity. The high-mannose variant of the RBD antigen (H-MAN/RBD) expressed in wild-type *Pichia pastoris* induced significantly higher IgG-specific antibody titers and pseudovirus neutralization activity compared with the complex RBD variant (Complex/RBD) expressed in mammalian cells (293F) or glycoengineering-capable *Pichia pastoris*. The rate of H-MAN/RBD adsorption onto aluminum hydroxide (alum) adjuvant was significantly higher than that of Complex/RBD. It was assumed that H-MAN/RBD might carry more negative charges because of its phosphomannose-modified surfaces, leading to a higher rate of adsorption onto the positively charged alum and enhancing the immune response. To assess the impact of phosphomannose modification on antigen immunogenicity, a yeast strain was engineered to prepare a low-mannose RBD antigen (L-MAN/RBD); additionally, a yeast strain was constructed to generate a low-phosphomannose-modified RBD antigen (L-MAN-P/RBD). In conclusion, phosphomannose modification substantially enhanced the immunogenicity of RBD by altering the surface charges of the RBD antigen and facilitating its adsorption onto alum. These findings offer novel insights and strategies for vaccine design and immunotherapeutic approaches.

## 1. Introduction

Severe acute respiratory syndrome coronavirus 2 (SARS-CoV-2) is the causative agent of coronavirus disease 2019 (COVID-19). Its rapid dissemination has posed unprecedented challenges to global healthcare systems, constituting a severe threat to human life and health on a global scale [1,2]. Vaccination represents the most efficacious strategy for halting viral transmission, and the development of SARS-CoV-2 vaccines has been a pressing priority [3,4]. Currently, subunit vaccines are preferred because of their cost-effectiveness, feasibility for large-scale production, and good safety and efficacy profiles. Research on vaccine adjuvants is crucial for enhancing the immunogenicity of subunit vaccines. Studies have shown that optimizing and adjusting the use of adjuvants can significantly improve the immune response. Aluminum hydroxide (alum) is the most commonly used adjuvant in the majority of subunit vaccines [5,6]; however, the efficacy of protein-based vaccines is closely associated with the binding between antigen and adjuvant, and the immunogenicities of SARS-CoV-2 subunit vaccines that use alum as their sole adjuvant are low [7,8,9]. Because of this, strategies for augmenting the immunogenicity of subunit vaccines are of critical importance.

Recent research on COVID-19 vaccines has predominantly centered on the full-length spike (S) protein and its receptor-binding domain (RBD). The RBD is a pivotal region of the S protein and harbors major neutralizing epitopes [10]. SARS-CoV-2 attaches to the human angiotensin-converting enzyme 2 (ACE2) receptor through the RBD, entering host cells and causing subsequent infection [11,12]. Glycosylation is a prevalent post-translational modification in eukaryotes. Among the diverse forms of protein glycosylation, N- and O-glycosylation are two of the most extensively investigated types [13,14]. N-glycosylation involves the attachment of oligosaccharides to asparagine residues. The SARS-CoV-2 S protein possesses 22 N-glycosylation sites, with approximately 40% of the protein surface being obscured by glycosylation. The RBD contains two critical N-glycosylation sites (N331 and N343) [15,16]. Watanabe et al. [17] identified that a variety of glycans play pivotal roles in viral self-protection and immune evasion, augmenting the virus’s ability to subvert innate immune surveillance by obscuring epitopes recognized by antibodies. Glycosylation can profoundly influence viral characteristics, potentially compromising the efficacies of currently available vaccines [18]. Crispin et al. [19] employed site-specific mass spectrometry to analyze the glycoform structures at the N-glycosylation sites of the recombinant SARS-CoV-2 S protein, which are significant for immune evasion, protein conformation, and cellular localization. N-glycosylation at the N343 site predominantly features complex polysaccharides containing abundant fucosylation and sialylation modifications. Increased levels of fucosylation and sialylation have been noted in the RBD of BA.5 and XBB.1 variants compared with the wild-type strain [20,21]. Mutations within the RBD region can influence the polysaccharide composition at the N343 site by modifying the mechanism of polysaccharide modification. Previous research on the glycosylation of the human immunodeficiency virus type 1 envelope glycoprotein (HIV-1 Env) has indicated that the processing of complex glycans is host cell-mediated. However, the extent of the glycosylation of oligosaccharides is largely independent of the expression system, and is more closely associated with protein structure and glycan density [22].

Our preliminary study revealed that the antibody titers elicited by the high-mannose RBD antigen (H-MAN/RBD) expressed in wild-type *Pichia pastoris* (*Komagataella phaffii*) were higher than those induced by the complex variant of RBD (293F/RBD) expressed in EXPi293F mammalian cells. To determine whether this immunogenicity difference stemmed from glycan structures, we used a complex glycoengineering yeast-derived RBD (Complex/RBD) to validate that this difference was caused by glycoforms [23]. Subsequent assessment of the adsorption rate of the antigen onto alum revealed that H-MAN/RBD was fully adsorbed, whereas the adsorption efficiency of Complex/RBD was lower. Mass spectrometry revealed that H-MAN/RBD possessed phosphomannose-modified N-glycans, whereas Complex/RBD lacked phosphomannose modifications. Based on early research into the interaction between phosphorylated proteins and alum and the regulation of the adsorption of antigens onto alum through precise charge regulation, we hypothesized that phosphomannose might influence immunogenicity. We hypothesized that phosphomannose residues enhance immunogenicity by facilitating antigen–adjuvant binding. To test this hypothesis, we genetically engineered yeast by knocking out the α-1,6-mannose transferase (*OCH1*) gene to produce an RBD antigen with low mannose modification (L-MAN/RBD) [24]. Additionally, we prepared a low-phosphomannose-modified RBD antigen (L-MAN-P/RBD) by further knocking out the phosphomannose transferase (*PNOIB*) and phosphomannose synthase (*Mnn4B*) genes in the yeast. Alum adsorption analysis demonstrated near-complete binding (>90%) of phosphomannose-rich antigens (H-MAN/RBD and L-MAN/RBD), while L-MAN-P/RBD showed significantly reduced adsorption (<10%). Critically, immunization of the mice revealed that phosphomannose-rich antigens (H-MAN/RBD and L-MAN/RBD) elicited higher antibody titers than L-MAN-P/RBD and Complex/RBD. This immunogenicity enhancement is directly mediated by the adsorption of the RBD promoted by phosphomannose onto alum.

## 2. Materials and Methods

### 2.1. Materials

*Pichia pastoris* strain, Expi293F, HEK293T, and HEK293T-ACE2 cells, each with distinct glycosylation capacities, were all maintained in our laboratory. The glycoengineered *Pichia pastoris* strains were constructed according to the method established by Choi et al. [25]. Reconstructed glycoengineered *Pichia pastoris* Glycoeng-yeast (Complex) had Δoch1, Δpno1, Δmnn4B, Δarm2, and *Kluyveromyces lactis* UDP-GlcNAc transporters; *Trichoderma reesei* α-1,2-MnsI; *Homo sapiens* (*H. sapiens*) β-1,2-GlcNAc transferase I; *Rattus norvegicus* β-1,2-GlcNAc transferase II; *Caenorhabditis elegans* MnsII; *Schizosaccharomyces pombe* Gal epimerase; *Drosophila melanogaster* UDP-Gal transporter; and *H. sapiens* β-1,4-galactosyltransferase. Reconstructed glycoengineered *Pichia pastoris* Glycoeng-yeast (L-MAN) had Δoch1, Δpmt1, and Δarm2. Reconstructed glycoengineered *Pichia pastoris* Glycoeng-yeast (L-MAN-P) had Δoch1, Δpmt1, ΔPNOIB, and Δmnn4B. *Escherichia coli* Trans5α competent cells were purchased from TransGen BioTech (Beijing, China). The pPICZαA-RBD plasmid and PCDNA3.1-RBD were constructed and stored at our laboratory. LiPofectamine™ 3000 transfection reagent was purchased from Invitrogen (Waltham, MA, USA). The SARS-CoV-2 RBD antibody was purchased from Sino Biological (Beijing, China). Horseradish peroxidase (HRP)-conjugated anti-His-tag monoclonal antibodies were obtained from Sigma (St. Louis, MA, USA), and HRP-conjugated goat anti-mouse IgG, IgG1, and IgG2a secondary antibodies were acquired from Abcam (Cambridge, UK). Restriction enzymes including XhoI, SacI-HF, BglII, and BamHI were obtained from New England Biolabs (Ipswich, MA, USA). The Bright-Lite™ Luciferase Assay System was purchased from Vazyme (Nanjing, China). Alum adjuvant was purchased from Croda (Frederikssund, Denmark). A TIANprep Mini Plasmid kit was acquired from Tiangen Biotech (Beijing, China). A ZS90 particle size potentiometer was acquired from Malvern Instruments Ltd. (Malvern, UK).

### 2.2. Cloning and Expression of SARS-CoV-2 RBD Genes in Yeast Cells

The gene encoding the SARS-CoV-2 RBD (GenBank: MN90908947.3; residues 216 and 319 to 534) was optimized based on the codon preference in *Pichia pastoris*. The gene was then inserted into the *Pichia pastoris* secretion expression vector pPICZαA, incorporating the XhoI and SalI restriction sites, and the resulting construct was designated as pPICZαA-RBD. The constructed pPICZαA-RBD plasmid was linearized at the BglII restriction site and subsequently electroporated into yeast strains engineered for generating complex glycoform, high-mannose-form, low-mannose-form, and low-phosphomannose-form proteins. In the four glycoengineering-capable competent yeast strains, expression was induced with 1% methanol. The target protein was detected by sodium dodecyl sulfate (SDS)-polyacrylamide gel electrophoresis (PAGE), followed by Coomassie blue staining and Western blotting. Engineered strains with high expression of target antigens were screened using HRP-conjugated anti-His antibody.

### 2.3. Cloning and Expression of SARS-CoV-2 RBD Genes in Mammalian Cells

The gene encoding the SARS-CoV-2 RBD was optimized according to mammalian codon usage bias and subsequently cloned into the expression vector pcDNA3.1 using *BamHI* and *XhoI* restriction sites. The resulting construct was designated as pcDNA3.1-RBD. EXPi293F cells were thawed and seeded into cell culture flasks, with a controlled cell density of 2 × 10^6^ to 5 × 10^6^ cells. The flasks were then placed in a CO_2_-filled incubator at 37 °C. When the cell confluency reached approximately 90%, the cells were routinely passaged. Transfection with Lipofectamine™ 3000 was performed when the cell density reached 1.2–1.7 × 10^6^ cells after three or more passages. In tube A, Lipofectamine 3000 was mixed with Opti-MEM™ at a ratio of 45 μL:750 μL. In tube B, the transfection reagent and plasmid were diluted in Opti-MEM™ at a ratio of 750 μL:7.5 μg:30 μL for Opti-MEM™, DNA, and P3000, respectively. The gently mixed solution from tube B was added into tube A and incubated at room temperature for 15 min. The mixture was then pipetted dropwise into 30 mL of Free Style™ 293 medium and incubated on a shaker at 37 °C for 72 h. The supernatant was collected by centrifugation.

### 2.4. Cultivation and Purification of the Recombinant RBD Protein in Shake Flasks

All cell lines expressing recombinant RBD proteins were scaled up in shake flasks, and the proteins were subsequently purified using Chelating FF nickel-column affinity chromatography. The supernatants and yeast cells were collected by centrifugation. After the pH was adjusted to 7.5, chromatography was performed using Mobile Phase A (20 mM Tris-HCl at pH 7.5, 500 mM NaCl, and 5 mM imidazole) and Phase B (20 mM Tris-HCl at pH 7.5, 500 mM NaCl, and 500 mM imidazole). After sample loading, the chromatography system was equilibrated to baseline for Phase A using ultraviolet detection. Elution was carried out sequentially with gradients of 5% B, 25% B, 50% B, and 100% B, followed by a final elution with NaOH. Elution peak samples were collected. Upon the completion of purification, the columns were preserved with a 20% ethanol solution. A 5% B solution was employed for the washing of heterologous proteins, while a 25% B solution was used to elute the target proteins. The elution samples containing the target proteins were subjected to ultrafiltration and concentration, followed by buffer exchange with 0.9% sodium chloride solution. The molecular weight and purity of the proteins were identified by SDS-PAGE.

### 2.5. Glycoform Identification

Protein samples were deglycosylated by the addition of an appropriate amount of PNGase F. Glycone was labeled with 2-AB fluorescence, and mass spectrometry was performed using a Waters ACQUITY UPLC I-Class liquid chromatography system and an Xevo G2 XS QTOF mass spectrometer. The precise masses of the oligosaccharides in the samples were determined by mass spectrometry, and the structural information and chromatograms of the oligosaccharides within the protein samples were elucidated through database searching and matching. This experiment was completed by Applied Protein Technology (Shanghai, China).

### 2.6. Mass Spectrometry for Measuring the Relative Molecular Masses (RMMs) of RBD Proteins

The RMMs of RBD proteins modified with various glycoforms were analyzed using a Bruker ultrafleXtreme MALDI-TOF/TOF mass spectrometer (Bruker Corporation, Billerica, Massachusetts, USA). After a 1 μL of sample was taken, 1 μL of 2,5-DHAP was added for MALDI RMM measurement. The raw data and spectra generated were subsequently analyzed using the Bruker Flex Analysis software (Flex Analysis version 3.4). This experiment was completed by Applied Protein Technology (Shanghai, China).

### 2.7. Measurement of the RBD Proteins’ Zeta Potential

The RBD proteins’ Zeta potential was measured using a DLS particle size potentiometer (Anhui Guoke Biotechnology Co., Ltd., Anhui, China). The RBD proteins were dissolved in phosphate buffer (0.01 mol/L, pH 7.0) to prepare a 0.1 mg/mL protein solution. The protein solution was fully hydrated and centrifuged at 12,000 r/min for 10 min. The supernatant was passed through a 0.45 μm Millipore filter (Merck KGaA, Darmstadt, Germany) and equilibrated at 25 °C for 30 s before testing.

### 2.8. Determination of the Level of Adsorption of the Recombinant RBD Antigen to Alum

A sterilized small beaker with spindles was placed on the magnetic stirrer of an ultra-clean table. Normal saline was added to the beaker, and the target protein solution (5 μg) was slowly added while stirring. Next, 100 μg of alum was added dropwise until it was completely dissolved. The mixture was then allowed to adsorb at 4 °C overnight. After adsorption overnight, the sample was centrifuged at 5000 rpm for 10 min. The supernatant was transferred to a new 1.5 mL EP tube. The centrifuged precipitate was resuspended in phosphate-buffered saline (PBS) containing 2% SDS and incubated at 37 °C for 30 min to facilitate desorption. Next, 80 μL of adsorption supernatant and 80 μL of adsorption precipitate were each added into 1.5 mL centrifuge tubes, to which 20 μL of 5× loading buffer was added. The samples were mixed well, heated in a bath of boiling water for 5 min, and centrifuged at 12,000 rpm for 1 min. The volume of sample loaded into each well was 40 μL. The experimental data were analyzed using SDS-PAGE, Western blotting, and grayscale analysis.

### 2.9. Grayscale Analysis

Western blots were imported and converted into grayscale images, with background interference removed for gray value calculation using Image J software (Image J v2.1.4.7). The data were processed using normalization methods and plotted using GraphPad Prism^®^ 8.

### 2.10. Animal Experiments

Seven-week-old SPF-grade BALB/c female mice, purchased from Beijing Vital River Laboratory Animal Technology Co., Ltd. (Beijing, China), were randomly grouped (*n* = 10 in each group) and housed in the Laboratory Animal Center of the Military Medical Research Institute of the Chinese People’s Liberation Army. All animal experiments were conducted in accordance with the guidelines of the Institutional Animal Ethics and Use Committee of the Beijing Institute of Bioengineering, and were approved by the committee (laboratory animal welfare ethics number: IACUC-DWZX-2020-039). The mice were immunized by intramuscular injection; the experimental group received a 5 μg dose of RBD protein along with 100 μg of alum, while the control group was administered 100 μg of alum alone, which was adsorbed overnight at 4 °C after preparation. The mice were immunized three times with an interval of 14 days. Blood samples were collected from the orbital venous sinuses before the first dose and 14 days after the second and third doses; after centrifugation, the supernatant was harvested and stored at −20 °C for later use. The titers of serum IgG, IgG1, IgG2b, IgG2a, and neutralizing antibodies were quantified.

### 2.11. Enzyme-Linked Immunosorbent Assay (ELISA)

ELISA was employed to quantify the titers of SARS-CoV-2 RBD-specific IgG, IgG1, IgG2a, and IgG2b antibodies in the sera of the immunized mice. The 96-well ELISA plates were pre-coated with the glycosyl-engineered yeast-expressed RBD glycoprotein (2 μg/mL) overnight at 4 °C. After washing the plates three times with phosphate-buffered saline with 0.1% Tween-20 (PBST), we blocked the plates with 5% skim milk (diluted in PBST) at 37 °C for 1 h. The serum samples obtained from the immunized mice were serially diluted in 5% skim milk and added to the ELISA plates. After incubation at 37 °C for 1 h, the plates were washed four times. Horseradish peroxidase (HRP)-conjugated goat anti-mouse IgG, IgG1, IgG2a, and IgG2b antibodies (dilution: 1:2500) were added to the ELISA plates, and the reaction was carried out at 37 °C for 1 h. Following four washes, 100 μL of 3,3′,5,5′-tetramethylbenzidine (TMB) single-component chromogenic solution was added to each well for color development, which lasted 5 min and was terminated by the addition of 50 μL of 2 mol/L H_2_SO_4_ to each well. The optical density (OD) value was measured at 450 nm using a microplate reader. The ELISA endpoint titer was defined as the highest serum dilution that gave a reading 2.1-fold greater than the background reading at 450 nm, and it was converted to logarithmic-scale before analysis.

### 2.12. Preparation of the SARS-CoV-2 Pseudovirus

The SARS-CoV-2 pseudovirus was prepared according to the method described by Liu et al. [26]. Well-grown HEK293T cells were passaged and then seeded onto Petri dishes one day in advance at a density of approximately 5 × 10^5^ cells/mL. After seeding, 12 mL of medium was added to each dish to reach a total cell count of approximately 6 × 10^6^. The cells were cultured in a 37 °C 5% CO_2_ incubator for 12–16 h. Transfection was performed when cells reached 70–90% confluence. The PCDNA-3.1-HIV-Luc backbone plasmid and the PCDNA-3.1-SARS-CoV-2-S plasmid were co-transfected using Lipofectamine™ 3000 Transfection Reagent. Reduced-serum medium was used following 6–8 h of transfection. After 72 additional hours of incubation, the cellular supernatant was harvested to obtain the SARS-CoV-2 pseudovirus, which was stored at −80 °C in 1.5 mL EP tubes.

### 2.13. Pseudovirus Neutralization Assay

The well-grown HEK293T-ACE2 cells were diluted to a density of 2 × 10^5^ cells/mL, and 100 μL of cells was added to each well to achieve a cell density of 2 × 10^4^ cells per well before culturing overnight. Subsequently, 10 μL of mouse serum was harvested and inactivated at 56 °C for 30 min. A cell control (CC) was prepared by adding 150 μL of medium, and a virus control (VC) was prepared by adding 100 μL of medium. The medium in the first well was diluted 25 times, and then a serial dilution was performed. Next, 50 μL of SARS-CoV-2 pseudovirus was added to each well, except the CC. The mixture of serum and pseudovirus was shaken for 30 s and incubated at 37 °C, in an atmosphere of 5% CO_2_, for 1 h. The medium from the HEK293T-ACE2 cells that had been cultured overnight was discarded, and 120 μL of the incubated mixture was transferred to the plate. After 8–12 h, DMEM medium was added, and the mixture was incubated at 37 °C and 5% CO_2_ for 48 h. Upon completion of the culture, 100 μL of supernatant was harvested, into which 100 μL of Bright-GLo luciferase detection reagent was added; the reaction was allowed to proceed at room temperature in the dark for 2 min. After repetitive pipetting, 100 μL of the reaction mixture was transferred to a white plate, and its relative luminescence units (RLUs) were then read using a microplate reader. The 50% effective concentration (EC50) value was calculated accordingly [27].

### 2.14. Statistical Analysis

The experimental data were processed using GraphPad Prism^®^8 and analyzed using *t*-tests and one-way analysis of variance. A *p* value < 0.05 was considered statistically significant.

## 3. Results

### 3.1. RBD Antigens Expressed by Wild-Type Pichia Pastoris Induced Higher Antibody Titers than Those Expressed in Mammalian Cells

RBD gene sequences containing two N-glycosylation sites (N331 and N343) were chosen for vector construction and protein expression. Figure 1a depicts S protein and RBD genes, while Figure 1b illustrates SDS-PAGE and Western blotting of RBD antigen proteins expressed in different systems. In immunized mice, the IgG-specific antibody titer induced by the H-MAN/RBD prepared using wild-type *Pichia pastoris* was 2.8 × 10^4^, significantly higher than the IgG-specific antibody titer (7.9 × 10^3^) induced by the RBD protein with complex glycosylation (293F/RBD) prepared in the mammalian cell expression system (*p* < 0.05) [28]. To explore whether the discrepancy in antibody titers was attributable to differences in glycoforms or expression systems, Complex/RBD expression in yeast was compared with 293F/RBD expression in mammalian cells. There was no significant difference in the antibody titers of Complex/RBD between yeast and mammalian cell systems (*p* > 0.05); additionally, the specific antibody titers induced by H-MAN/RBD were significantly higher than those induced by yeast-expressed Complex/RBD (5.6 × 10^3^) (*p* < 0.01) (Figure 1c). The neutralizing antibody titer induced by H-MAN/RBD in immunized mice was 1:123, which was significantly higher than that induced by Complex/RBD (1:26) or 293F/RBD (1:30) (*p* < 0.05) (Figure 1d). The IgG-specific and neutralizing antibody titers elicited by H-MAN/RBD expressed by wild-type *Pichia pastoris* were significantly higher than those induced by the Complex/RBD antigen.

Because proper glycosylation is critical for maintaining protein structure and function, the glycoforms of the above RBD proteins were precisely analyzed using high-resolution mass spectrometry and ultra-high-performance liquid chromatography. The N-glycans of the H-MAN/RBD antigen predominantly consisted of M9 and M10, accounting for 14.8% and 17.1% of the total glycan population, respectively (Appendix A). In addition, many phosphomannose-modified N-glycans were found (Figure 1e). For the Complex/RBD antigen, the prevalent N-glycans were the biantennary complex glycoforms Gal2Man3GlcNAc2 and M5 (Figure 1f), comprising 54.4% and 27.56% of the total glycans, respectively (Appendix A). For the 293F/RBD antigen, the N-glycans were also predominantly biantennary complex glycoforms (Figure 1g), with no phosphomannose modification detected (Appendix A). It was assumed that the higher antibody titers elicited by H-MAN/RBD expressed by wild-type *Pichia pastoris* compared with those induced by Complex/RBD might be attributed to the impact of phosphomannose on immunogenicity.

### 3.2. Reduced Construction of the Phosphate Mannose RBD Antigen by PNOIB and the MNN4B Knockout Yeast Strain, and Verification of Its Glycosylation

Low-phosphate mannose-type RBD antigens were constructed to further elucidate the impact of high mannose and phosphomannose on the immunogenicity of RBD antigens. The RBD antigen was first expressed using a yeast strain engineered to exhibit a reduced mannosylation pathway, characterized by the knockout of the *OCH1* gene [24]. Subsequently, a yeast strain with diminished phosphomannose modification was constructed by knocking out the phosphomannose transferase (*PNOIB*) and phosphomannose synthase (*MNN4B*) genes in the background of the previously engineered low-mannose yeast strain (Figure 2a). The knockout plasmids were randomly integrated into the host chromosomes, and repetitive sequences appeared in the yeast after integration. Homologous exchanges occurred between two copies in the yeast chromosomes, facilitating exchanges of the 5’ and 3’ homology arms and the lox-zeocin sequences of the *PNOIB* gene with the corresponding regions of the *PNOIB* gene’s open reading frame (ORF), which enabled the knockout of the *PNOIB* gene through a single recombination. The recombinant strains encompassed either wild or *PNOIB* knockout variants. Strains that failed to effectively knock out the *PNOIB* gene lacked Zeocin resistance and consequently were unable to proliferate on YPD plates supplemented with Zeocin. Following the successful construction of the *PNOIB* knockout strain, the *Mnn4B* gene was subsequently knocked out using a similar approach, generating a low-phosphomannose yeast strain.

The low-mannose (L-MAN/RBD) and low-phosphomannose (L-MAN-P/RBD) RBD antigens were successfully expressed in the engineered low-mannose and low-phosphomannose yeast strains. Subsequently, the target proteins were purified using nickel-column affinity chromatography. Subsequent analysis of the glycoforms of the L-MAN/RBD and L-MAN-P/RBD antigens (Figure 2b,c) revealed that the N-glycans of L-MAN/RBD predominantly comprised M10_Phosphorylation2 and M11_Phosphorylation2. In contrast, the N-glycans of the L-MAN-P/RBD antigens were primarily composed of M5 and M6. The N-glycans modified by mannose phosphate in the L-MAN/RBD antigen represented 79.95% of the total glycans; in contrast, the N-glycans modified by mannose phosphate in the L-MAN-P/RBD antigen constituted 28.32% of the total glycans. There was a 51.63% reduction in the proportion of mannose phosphate-modified N-glycans compared with those in L-MAN-P/RBD expressed in low-mannose yeast strains (Appendix A), indicating that the de-phosphomannose-modified L-MAN-P/RBD antigen was successfully obtained. The molecular weight of the RBD protein, modified by various glycoforms, was initially assessed using SDS-PAGE and Western blotting (Figure 2d). Differences in the molecular weight of different glycoforms of the RBD obtained by purification were seen, with the order being Complex < L-MAN-P < L-MAN < 293F < H-MAN. Following digestion with the N-glycosidase PNGase F (Figure 2e), the molecular weights were reduced to the theoretical weight of approximately 25 kDa, confirming the presence of varying degrees of glycosylation.

### 3.3. Phosphomannose Modification Enhanced the Adsorption of RBD Antigens onto Alum

We compared the adsorption levels of RBD antigens with various glycoforms onto alum (Figure 3). The H-MAN/RBD antigen exhibited a high adsorption level (93%) (Figure 3a). In the Complex/RBD vaccine, around 56% of the RBD antigen remained in the supernatant, with only 35% adsorbed onto alum. (Figure 3b) The adsorption efficiency of the L-MAN/RBD vaccine was comparable to that of the H-MAN/RBD vaccine (94%) (Figure 3c). In contrast, the adsorption level of the L-MAN-P/RBD antigen, expressed by the phosphomannose knockout host strain, differed from the other three vaccine groups; approximately 85% of the antigen was found in the supernatant, while only 10% adsorbed onto alum (Figure 3d). The adsorption level of RBD antigens with different glycoform modifications onto alum was L-MAN-P < Complex < 293F < H-MAN < L-MAN. The H-MAN/RBD and L-MAN/RBD antigens, which had a large number of phosphomannose modifications, exhibited a strong adsorption efficiency; in contrast, the adsorption level of the L-MAN-P/RBD antigen, expressed by the phosphomannose knockout yeast system, had significantly reduced adsorption efficiency. When different glycoform RBD antigens were digested with PNGFase, the adsorption efficiency of H-MAN and L-MAN onto alum decreased, and was basically similar to that of L-MAN-P and Complex (Appendix A). The Zeta potentials of H-MAN/RBD and L-MAN/RBD were both negative and similar, mainly due to the modification of phosphomannose modification. The negative charge of L-MAN-P/RBD was reduced because there was no phosphate group modification (Appendix A). Overall, phosphomannose promoted the adsorption of RBD antigens onto alum, thereby improving vaccine immunogenicity.

### 3.4. Higher Antibody Titers Elicited by RBD Antigens Expressed by Wild-Type Pichia Pastoris Were Correlated with Phosphomannose Modification

To investigate the impact of RBD antigens with various glycoforms on immunogenicity, we chose alum, known for its favorable safety profile, as an adjuvant for immunizing BALB/c mice (Figure 4). Specific antibodies were detected following immunization (Figure 4a). The antibody titers were 4.5 ± 0.21 × 10^4^ in the H-MAN group and 3.2 ± 0.15 × 10^4^ in the L-MAN group (*p* = 0.5585). In contrast, the antibody titers were significantly lower in the L-MAN-P group (3.5 ± 0.09 × 10^3^) than in both the H-MAN and L-MAN groups (*p* < 0.001). The IgG antibody isotypes were further assessed after the second immunization; the serum IgG1, IgG2a, and IgG2b antibody titers are presented in Figure 4b, 4c, and 4d, respectively. The titers of IgG1 antibody were elevated in both groups (6.8 ± 0.25 × 10^4^ with H-MAN and 8.3 ± 0.11 × 10^4^ in L-MAN), with no significant difference between the two (*p* = 0.7414). The IgG1 antibody titer in the L-MAN-P group was 3.5 ± 0.19 × 10^3^, significantly lower than that in the H-MAN group (*p* < 0.01). Additionally, the IgG1 antibody titer in the Complex group was 1.3 ± 0.16 × 10^4^, which was significantly different from that in both the H-MAN and L-MAN groups (*p* < 0.05). The IgG2a antibody titer in the H-MAN group was 6.5 ± 0.28 × 10^2^, which was significantly lower than that in the L-MAN-P group (0.4 ± 0.12 × 10^2^; *p* < 0.001); additionally, the IgG2a antibody titer in the Complex group was 2.6 ± 0.17 × 10^2^, which was also significantly different from that in the H-MAN group (*p* < 0.05). The IgG2b antibody titer was 1.1 ± 0.23 × 10^3^ in the H-MAN group, significantly higher than that in the L-MAN-P group (1.2 ± 0.19 × 10^2^; *p* < 0.01). No significant differences in IgG2b antibody titers were observed among the other groups. IgG1 and IgG2a antibody isotyping demonstrated that the RBD subunit vaccines were capable of eliciting specific Th1 and Th2 immune responses against the RBD antigen.

Following immunization, the neutralizing antibody titer was 1:71 in the H-MAN group and 1:58 in the L-MAN group, showing no significant difference (*p* = 0.7420). In contrast, the neutralizing antibody titer of the H-MAN group was significantly higher than those of the L-MAN-P (1:25) and the Complex groups (*p* < 0.05) (Figure 4e). The antibody titers elicited by the RBD antigens modified with phosphomannose (H-MAN/RBD and L-MAN/RBD) were higher than those induced by the low-phosphomannose variants (L-MAN-P/RBD and Complex/RBD), indicating that phosphomannose enhanced the immunogenicity of RBD antigens. The changes in mice’s body weight within 1 week following the primary immunization were monitored and the weight gain curves were plotted (Figure 4f) to evaluate the safety of the recombinant RBD vaccines. Weight changes were consistent among different groups, and no abnormalities such as hair loss, lethargy, or fever were observed.

## 4. Discussion and Conclusions

Our present study found that the titers of IgG-specific and neutralizing antibodies elicited by H-MAN/RBD expressed by wild-type *Pichia pastoris* were significantly higher than those induced by the complex glycosylation RBD antigens expressed in mammalian cells and glycoengineering-capable yeasts. It was hypothesized that the enhanced immunogenicity of H-MAN/RBD was associated with its glycoforms, which were then further characterized. The N-glycans of the H-MAN/RBD antigens expressed in wild-type *Pichia pastoris* predominantly consisted of M9, M10, and phosphorylated M10_Phosphorylation, whereas the Complex/RBD antigens were primarily composed of the biantennary complex glycoforms Gal2Man3GlcNAc2 and M5; the key distinction between them was that the complex glycoforms lacked phosphorylated glycans.

Variations in glycoforms may contribute to differences in immunogenicity. To validate this hypothesis, we expressed and purified both low-mannose and low-phosphomannose RBD antigens (L-MAN/RBD and L-MAN-P/RBD). The antibody titers induced by H-MAN/RBD and L-MAN/RBD were higher than those induced by Complex/RBD and L-MAN-P, suggesting that the variations elicited by different glycoforms were associated with phosphomannose modification. Significant differences in RBD antigen adsorption rates onto alum were observed across various glycoforms; approximately 90% of the H-MAN/RBD and L-MAN/RBD antigens adsorbed onto alum, whereas only about 8% of the L-MAN-P/RBD antigens adsorbed. These results demonstrate that phosphomannose modification enhances the adsorption of RBD antigens onto the positively charged alum by increasing the negative charges on the antigen surface, enhancing its immunogenicity.

In a recent study, RBD antigens were conjugated with in vitro synthesized phosphoserine (pSer) peptides at the N terminus to generate pSer_4_-RBD or RBD-pSer_4_ that could efficiently adsorb onto the surface of alum particles, increasing the RBD-to-alum adsorption rate. This modification aided in slowing antigen clearance and markedly elevated the titers of neutralizing antibodies in mice. Leveraging ligand exchange mechanisms to enhance the binding of antigens onto alum could increase the efficacy of SARS-CoV-2 subunit vaccines [29]. Additionally, a novel strategy for enhancing the design of protein subunit vaccines formulated with alum has been proposed. By incorporating negatively charged amino acid residues such as aspartic acid (Asp) into the flexible regions of the SARS-CoV-2 RBD, precise modification of the protein surface can be accomplished; this strategy can alter the strength and orientation of antigen–alum interactions, facilitating robust adsorption of the antigen onto alum and impacting vaccine efficacy [30]. However, both methods necessitate the incorporation of additional amino acids to bolster the adsorption capacity of RBD antigens onto alum. Studies have shown that GAP112 modified by a hydroxylamine group or small-molecule TLR7/8 can be conjugated with the N-terminus of the RBD to construct an adjuvant–RBD conjugate that constitutes a biosafe vaccine. Compared to the unconjugated mixture of adjuvant and RBD, a two-dose immunization of the adjuvant–RBD conjugate vaccine strongly activated innate immune cells [5,6,31]. SARS-CoV-2 RBD is not sufficiently immunogenic without an adjuvant. SARS-CoV-2 RBD-based subunit vaccines can be designed by linking two or more RBDs from the same or different virus strains, or RBD can be fused to other immunogens. Similarly to the RBD single-chain dimers (sc-dimers) of MERS-CoV and SARS-CoV, a tandem repeat RBD-sc-dimer of SARS-CoV-2 elicits an increase in neutralizing antibodies, in contrast to an RBD monomer. Indeed, a triple RBD vaccine (3Ro-NC) comprising one Delta variant RBD and two RBDs from the Omicron-BA.1 subvariant elicits mucosal immune responses and the production of neutralizing antibodies against the Omicron-BA.1 subvariant, thereby protecting mice from Omicron infection and virus-induced immunopathology [32,33]. We directly expressed RBD antigens through glycoengineering-capable yeast in order to modify it with phosphomannose, which increased its adsorption rate onto alum and offers a novel approach to enhancing vaccine immune response.

Although phosphate groups can modulate the extent of antigen adsorption onto alum, the relationship between the extent of adsorption and immunogenicity is extremely intricate. Another study investigated the relationship between the adsorptive coefficient and immunopotentiation, and discovered that the strength of ligand exchange (adsorptive coefficient) was influenced by the number of phosphate groups on the antigen, as well as the number of surface hydroxyls on the adjuvant. Four vaccines were prepared, in which the adsorptive coefficient was varied by altering the number of phosphate groups on the antigen (alpha casein and dephosphorylated alpha casein) or the number of surface hydroxyls on the adjuvant (aluminum hydroxide and phosphate-treated aluminum hydroxide adjuvants). Mice immunized with the vaccine containing de-phosphorylated α-casein and phosphate-treated alum produced a higher antibody response; this vaccine also had the highest adsorptive coefficient, indicating that the antibody titer generated by the vaccine correlated positively with the adsorptive coefficient. However, T-cell activation was not observed in mice that received the vaccine with the highest adsorptive coefficient (alpha casein/aluminum hydroxide adjuvant), suggesting that antigen processing and presentation to T-cells is impaired when the antigen is adsorbed too strongly [34]. The above study underscores the importance of regulating the adsorptive coefficient in optimizing vaccine formulations for robust immune responses, while also navigating the potential drawbacks associated with excessive adsorption. The intensity of antibody responses induced by antigens is closely linked to the rate of antigen-to-alum adsorption. However, the extent of adsorption must be appropriate, neither excessively strong nor weak, in order to achieve optimal immunogenicity. Excessively strong adsorption may result in antigens being inadequately released after vaccination, impairing their full engagement with the immune system and diminishing the immune response. Conversely, if the adsorption rate is too low, antigens may be released prematurely or at insufficient local concentrations that cannot sustainably stimulate the immune system to generate an adequate immune response. This parameter must be meticulously considered and managed during vaccine development. A balanced adsorption rate should be identified to ensure that a vaccine releases the antigen with optimal timing and in an optimal quantity, triggering the immune system to generate adequate antibodies and cellular immune responses.

At present, glycoform determination has limitations. The glycoform analysis for H-MAN/RBD expressed by the wild-type Pichia pastoris system can only detect mannose structures up to mannose-1,3-α-D-mannose (M13) as the longest detectable structure, and is unable to detect longer mannose chains (such as M14, M15, and beyond). As a result, our analysis could not provide a complete picture of the H-MAN/RBD glycan profile. The optimization or development of analytical techniques with enhanced resolution and sensitivity for complex high-mannose glycans is therefore essential in future work.

## Figures and Tables

**Figure 1 biomolecules-15-01172-f001:**
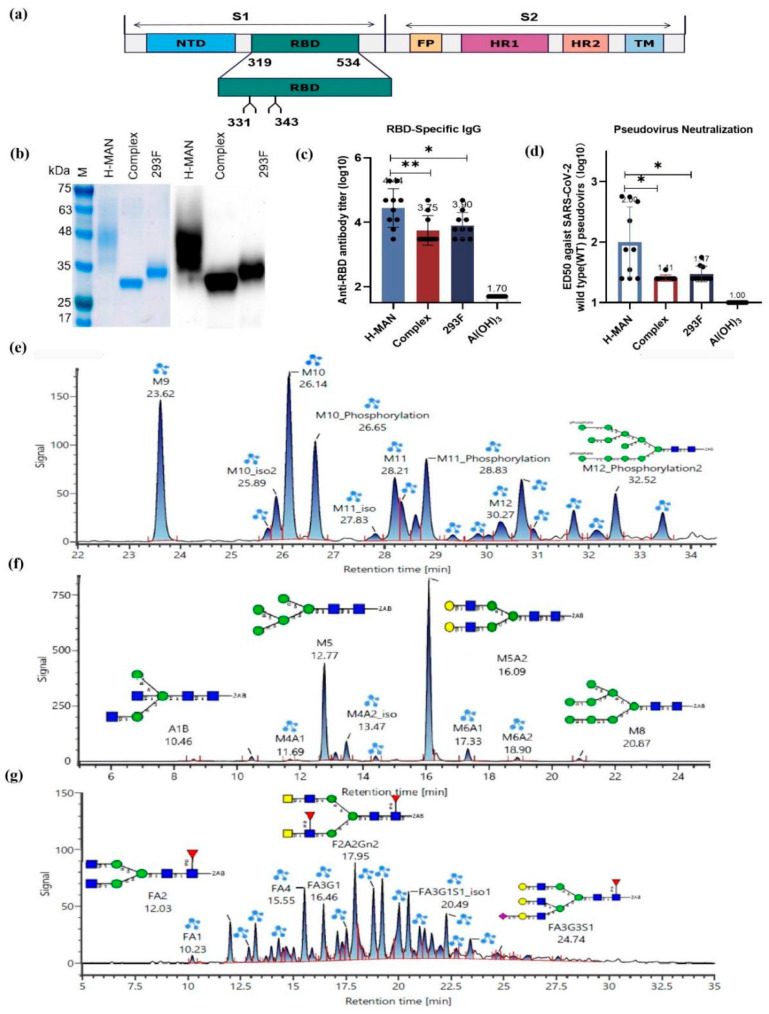
A schematic diagram of the construction, expression, glycoform identification, and immunogenicity analysis of different SARS-CoV-2 S-RBD proteins. (**a**) SARS-CoV-2 protein sequences. (**b**) Quantitative SDS-PAGE validation and Western blotting of RBD proteins (0.2 μg) expressed in three different vectors. (**c**) Detection of RBD-specific IgG antibodies in mouse serum samples following booster immunization. (**d**) Evaluation of neutralizing antibodies against SARS-CoV-2 pseudovirus in mouse serum after booster immunization (*t*-test; * *p* < 0.05, ** *p* < 0.01). (**e**) Fluorescent spectra of H-MAN/RBD antigen oligosaccharides. (**f**) Fluorescent spectra of Complex/RBD antigen oligosaccharides. (**g**) Fluorescent spectra of 293F/RBD antigen oligosaccharides.

**Figure 2 biomolecules-15-01172-f002:**
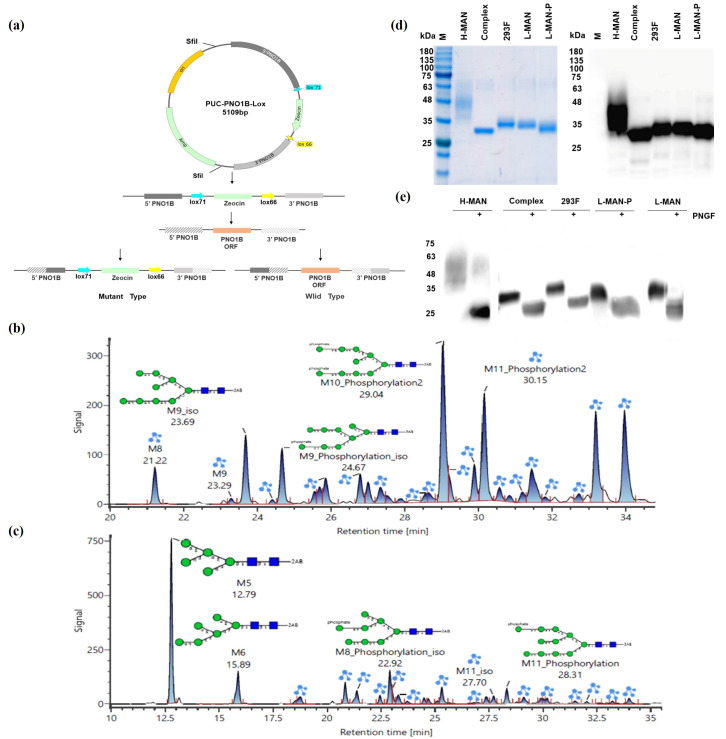
The expression of RBD antigens with different glycoforms and their identification. (**a**) A schematic diagram illustrating the knockout of target genes using homologous recombination. (**b**) Fluorescent spectra of L-MAN/RBD antigen oligosaccharides. (**c**) Fluorescent spectra of L-MAN-P/RBD antigen oligosaccharides. (**d**) Identification of RBD antigens with different glycoforms by SDS-PAGE and Western blotting. (**e**) Identification of RBD antigens with different glycoforms by restriction digestion using PNGF.

**Figure 3 biomolecules-15-01172-f003:**
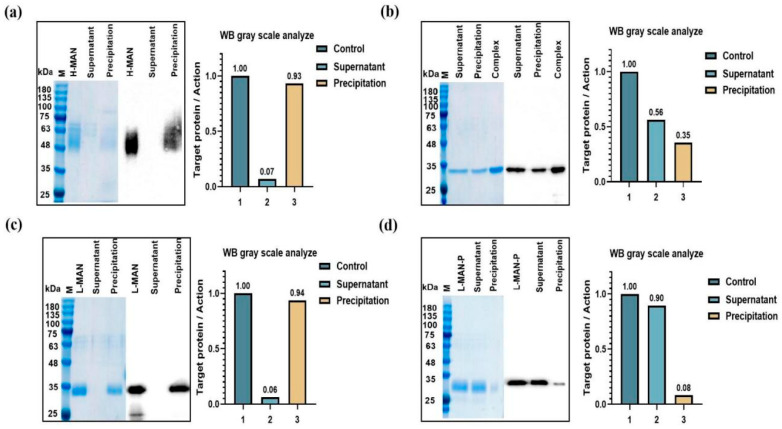
Determination of the adsorption levels of RBD antigens with different glycoforms onto aluminum hydroxide (alum). (**a**) The adsorption rate of the H-MAN/RBD antigen onto alum. (**b**) The adsorption rate of Complex/RBD antigen onto alum. (**c**) The adsorption rate of L-MAN/RBD antigen onto alum. (**d**) The adsorption rate of L-MAN-P/RBD antigen onto alum.

**Figure 4 biomolecules-15-01172-f004:**
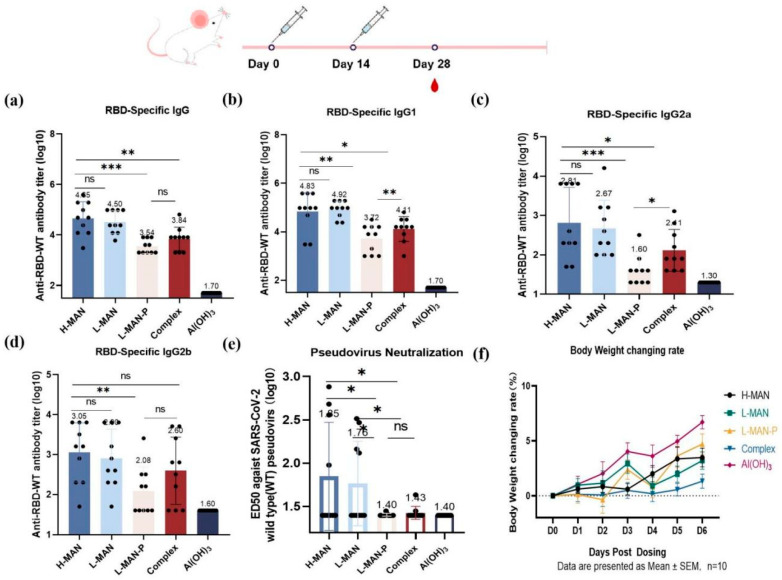
Assessment of humoral immune responses in mice immunized with RBD antigens with different glycoforms. (**a**) Titers of RBD-specific IgG antibodies in mouse serum. (**b**) Titers of RBD-specific IgG1 antibodies in mouse serum. (**c**) Titers of RBD-specific IgG2a antibodies in mouse serum. (**d**) Titers of RBD-specific IgG2b antibodies in mouse serum. (**e**) Titers of neutralizing antibody against SARS-CoV-2 pseudovirus in mouse serum. (**f**) Changes in body weight in each group. * *p* < 0.05; ** *p* < 0.01; ** *p* < 0.001; ns, not significant.

## Data Availability

The original contributions presented in this study are included in the article/Appendix A. Further inquiries can be directed to the corresponding authors.

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
