# Peer review of "Glycosylated SARS-CoV-2 RBD Antigens Expressed in Glycoengineered Yeast Induce Strong Immune Responses Through High Antigen–Alum Adsorption"

_biomolecules, 2025, doi:10.3390/biom15081172_

Round 1

Reviewer 1 Report

Comments and Suggestions for Authors

In the study by Li et al., the production of the receptor-binding domain (RBD) of the SARS-CoV-2 spike (S) protein with different glycosylation patterns was described. It was shown that immunization of mice with a high-mannose variant of the RBD antigen (H-MAN/RBD), expressed in the wild-type Komagataella phaffii (formerly Pichia pastoris), induced significantly higher titers of IgG-specific antibodies and pseudovirus neutralization activity compared to the complex-type RBD (Complex/RBD) expressed in mammalian cells (293F) or in K. phaffii with modified glycosylation. The study demonstrated that the glycosylation type influences the adsorption efficiency onto aluminum-based adjuvants, with H-MAN/RBD showing the highest adsorption. The authors proposed that H-MAN/RBD may carry more negative charges due to phosphomannose modifications on its surface, leading to stronger adsorption onto the positively charged aluminum adjuvant and enhanced immune responses.

To assess the role of phosphomannosylation in antigen immunogenicity, a yeast strain was engineered to produce a low-mannose RBD antigen (L-MAN/RBD), and another strain was constructed to generate an RBD antigen with reduced phosphomannose modification (L-MAN-P/RBD). The results confirmed that phosphomannosylation significantly enhances RBD immunogenicity by altering the antigen’s surface charge and promoting its adsorption onto the aluminum adjuvant.

The paper is written in a scientific style, includes a detailed introduction, and features a well-structured discussion. However, the Materials and Methods section requires expansion.

Comments:

  1. The correct species name, Komagataella phaffii (formerly Pichia pastoris), should be used in the Introduction.
  2. Line 57: Duplicate references should be removed.
  3. Materials and Methods:
    • Describe the construction of the pcDNA3.1-RBD plasmid.
    • Explain how the K. phaffii knockout strains were generated. This information must be added to the Materials and Methods section.
  4. Section 3.2: The heading should be revised for clarity.

Author Response

Dear Reviewers,

We have made substantial revisions to the manuscript “Glycosylated SARS-CoV-2 RBD antigens expressed in glycoengineered yeast induce strong immune responses through high antigen-alum absorption”(Manuscript Number: biomolecules-3713016)based on the comments raised by the four reviewers. The main textual changes made to the manuscript based on answering these comments are colored red in the manuscript.

Point-by-point responses to the comments raised by the reviewers are detailed below.

-Reviewer 

  1. The correct species name, Komagataella phaffii (formerly Pichia pastoris), should be used in the Introduction

Response:

Thank you for your suggestion. The new classification name Komagataella phaffii has been annotated in "Introduction" on line 99.

  1. Line 57: Duplicate references should be removed

Response:

Thank you for your comment. The duplicate reference has been removed on line 59.

  1. Materials and Methods:
  • Describe the construction of the pcDNA3.1-RBD plasmid.
  • Explain how the K. phaffii knockout strains were generated. This information must be added to the Materials and Methods section

Response:

Thank you to the reviewer for careful review and insightful comments. We added the description of pcDNA3.1-RBD plasmid construction on lines 162-165 "The gene encoding the SARS-CoV-2 RBD was optimized according to mammalian codon usage bias and subsequently cloned into the expression vector pcDNA3.1 using BamHI and XhoI restriction sites. The resulting construct was designated as pcDNA3.1-RBD". The glycoengineered Pichia pastoris strains were constructed according to the method established by Choi et al. Reconstructed glycoengineered Pichia pastoris Glycoeng-yeast(Complex) had Δoch1, Δpno1, Δmnn4B, Δarm2 and Kluyveromyces lactis UDP-GlcNAc transporters, Trichoderma reesei α-1,2-MnsI, Homo sapiens (H. sapiens) β-1,2-GlcNAc transferase I, Rattus norvegicus β-1,2-GlcNAc transferase II, Caenorhabditis elegans MnsII, Schizosaccharomyces pombe Gal epimerase, Drosophila melanogaster UDP-Gal transporter and H. sapiens β-1,4-galactosyltransferase. Reconstructed glycoengineered Pichia pastoris Glycoeng-yeast(L-MAN) had Δoch1, Δpmt1 and Δarm2. Reconstructed glycoengineered Pichia pastoris Glycoeng-yeast(L-MAN-P) had Δoch1, Δpmt1, ΔPNOIB and ΔMNN4B. The above content has been added in Materials and Methods section (page 4, lines 126-135) and colored red.

  1. Section 3.2: The heading should be revised for clarity.

Response:

Thank you for your suggestion. Section 3.2 mainly describes the construction of RBD antigens for the production of less mannose (L-MAN/RBD) and low phosphate mannose (L-MAN-P/RBD) by knocking out (PNOIB and MNN4B genes) on the basis of engineered yeast strains. The glycoform analysis experiment and SDS-PAGE/Western blot experiment confirmed that the molecular weight difference of RBD antigen was caused by glycosylation. Therefore, the title was changed to "construction of less phosphate mannose RBD antigen by PNOIB and MNN4B knockout yeast strain and its glycosylation verification".

Reviewer 2 Report

Comments and Suggestions for Authors

In this article, the authors sought to investigate the role of glycosylation in determining immunogenicity of alum-loaded SARS-CoV-2 S protein receptor-binding domain (RBD).  The central hypothesis is that phosphomannose modifications enhance antigen-alum binding via increased negative surface charge, thereby improving immune responses. The authors systematically compare RBD antigens with different glycoforms (high-mannose, complex, and engineered low-phosphomannose variants) to validate this hypothesis.  They found that high mannose (H-MAN) RBD was more effective at generating specific IgG antibodies and neutralizing activity in mouse sera than the complex isolated from 293T human cells. The presumption was that more phosphomannose on H-MAN from yeast gave a more negative charge, facilitating binding to alum adjuvant. Testing two genetic yeast mutants that had lower mannose but still relatively high phosphomannose (L-MAN), and a mutant lacking phosphomannose (L-MAN-P), the authors could demonstrate that phosphomannose indeed influences binding to alum, specific IgG antibody titers, and serum neutralizing activity.  The methodology is robust and well-described, employing standard techniques (e.g., SDS-PAGE, mass spectrometry, ELISA, pseudovirus neutralization assays) to characterize glycoforms and immune responses.  Overall, the work is thorough going from biochemical analysis by mass spectrometry and in vivo effects.  The experiments are described and visualized well, and are routine methodology of the field.  Overall, this work provides a useful consideration in developing effective vaccine candidates and advances the understanding of glycoform-adjuvant interactions. The findings could inform strategies for other pathogens.  

The topic is highly relevant to vaccine development, particularly for SARS-CoV-2 subunit vaccines. Glycosylation’s role in immunogenicity is well-documented, but this study provides novel insights into how specific glycoforms (e.g., phosphomannose) can optimize antigen-adjuvant interactions. The work addresses a gap in understanding how glycoengineering can be leveraged to enhance alum-based vaccines. One minor area is the lack of discussion on why L-MAN/RBD (low-mannose but high phosphomannose) performed similarly to H-MAN/RBD despite differing glycan sizes. Clarifying this would strengthen the argument for phosphomannose as the key determinant.  Compared to prior work focusing on glycosylation’s role in immune evasion or protein stability, this study directly links glycoform-specific properties (e.g., charge) to adjuvant compatibility and immunogenicity.

I have only a few minor text edits:
Highlights section- italicize P. pastoris throughout
Lines 64, 67: Remove the bolded error message
Introduction and Discussion- could the references be combined into a single bracket?  eg, [8][9] becomes [8-9].  The reference on line 57 is doubled
Line 63: insert a space after the period
Line 132: place a space before the parentheses
Line 157: perhaps viability is not the right term, but confluency
Lines 346 and 347: the formatting for PNOLB is different than other places in this paragraph
Check the formatting of line 429, perhaps a tab at the end to make the spacing between words consistent.
Line 496: there will need to be a space before "We"
Tables 4 and 5 in the supplement need a space before the number in the title. eg, Table4 to Table 4

Author Response

Dear Reviewers,

We have made substantial revisions to the manuscript “Glycosylated SARS-CoV-2 RBD antigens expressed in glycoengineered yeast induce strong immune responses through high antigen-alum absorption”(Manuscript Number: biomolecules-3713016)based on the comments raised by the four reviewers. The main textual changes made to the manuscript based on answering these comments are colored red in the manuscript.

Point-by-point responses to the comments raised by the reviewers are detailed below.

-Reviewer 2

  1. Highlights section- italicize P. pastoris throughoutï¼›Introduction and Discussion- could the references be combined into a single bracket?  eg, [8][9] becomes [8-9].  The reference on line 57 is doubledï¼›Line 63: insert a space after the periodï¼›Line 132: place a space before the parenthesesï¼›Line 496: there will need to be a space before "We"ï¼›Tables 4 and 5 in the supplement need a space before the number in the title. eg, Table4 to Table 4ï¼›Check the formatting of line 429, perhaps a tab at the end to make the spacing between words consistent.]

Response:

We greatly appreciate your valuable comments regarding formatting consistency. All suggested revisions have been carefully addressed as follows: "P. pastoris has been italicized throughout the Highlights sectionï¼›The format of references involved in the full text has been changedï¼›The question of spaces has also been examined in full text. " Additionally, we have conducted a full-text audit to verify. The revised manuscript with tracked changes is attached for your convenience. We sincerely thank you for enhancing the precision of our work.

  1. Lines 346 and 347: the formatting for PNOLB is different than other places in this paragraph]

Response:

We sincerely thank the reviewer for highlighting this inconsistency. The following corrections have been implemented. All instances of the phosphomannose transferase gene designation have been standardized to "PNOLB" (previously appearing as  "PNOI.B" in Lines115\346\347...).

  1. Lines 64, 67: Remove the bolded error messageï¼›Line 157: perhaps viability is not the right term, but confluenc

Response:

Thank you for your suggestion. The bolded error messages have been removed. The " viability " was modified to " confluency " on line 167.

Reviewer 3 Report

Comments and Suggestions for Authors

Dear Dr. Bo Liu and Dr. Jun Wu,

Thank you for submitting your manuscript to Biomolecules. I have conducted a comprehensive review of your work investigating the relationship between glycosylation patterns and immunogenicity of SARS-CoV-2 RBD antigens. While your glycoengineering approach aligns well with Biomolecules' focus on bioactive substances and molecular mechanisms, I must recommend major revisions due to significant methodological gaps and concerns regarding the manuscript's contemporary relevance within the journal's scope.

MANUSCRIPT ALIGNMENT WITH BIOMOLECULES SCOPE

Your research fits within the journal's mission focusing on "structures and functions of bioactive and biogenic substances" and "molecular mechanisms with biological and medical implications." The glycoengineering platform you describe represents an interesting biomolecular modification approach. However, the execution requires substantial improvement to meet the standards expected for a Q1 journal with Impact Factor 4.8.

SUMMARY OF FINDINGS

You demonstrate that phosphomannose-modified RBD variants (H-MAN/RBD, L-MAN/RBD) achieve superior aluminum hydroxide adsorption (>90% vs ~35% for complex glycoforms) and generate enhanced antibody responses in BALB/c mice. Your mass spectrometry characterization effectively demonstrates distinct N-glycan profiles, with phosphomannose modifications correlating with improved adjuvant binding through proposed electrostatic interactions.

MAJOR SCIENTIFIC DEFICIENCIES

1. Methodological Rigor Below Journal Standards

Statistical Design Inadequacies:

  • Sample size (n=10) is insufficient for immunological studies in a Q1 journal. Biomolecules standards require n≥15-20 with appropriate power calculations.
  • Absence of multiple comparison corrections despite extensive pairwise testing violates statistical best practices.
  • No a priori power analysis provided to justify experimental design.

Incomplete Molecular Mechanism Characterization:

  • Your zeta potential data (-3.67 vs -3.69 mV) shows differences within measurement error, insufficient to support the proposed charge-based mechanism.
  • Missing quantitative phosphomannose content determination - crucial for a journal emphasizing molecular characterization.
  • Acknowledged limitations in glycan analysis (>M13 detection) compromise the biomolecular characterization central to this journal's scope.

2. Insufficient Immunological Assessment for Medical Implications

Critical Gap in Cellular Immunity: For a manuscript claiming "strong immune responses" in a journal covering "biological and medical implications," you must include:

  • CD4+/CD8+ T-cell response analysis
  • Cytokine profiling (IFN-γ, IL-2, IL-4, IL-17)
  • Memory immune response characterization
  • Th1/Th2 polarization assessment

Inadequate Functional Validation:

  • Neutralization titers (EC50 1:123) are below established protective thresholds
  • No in vivo protection studies to validate biological relevance
  • Limited to humoral responses without comprehensive immune system evaluation

3. Molecular Characterization Gaps

Incomplete Structure-Function Analysis:

  • Missing detailed thermodynamic analysis of glycan-adjuvant interactions
  • No desorption kinetics data to understand sustained antigen presentation
  • Insufficient characterization of the molecular basis for enhanced immunogenicity claims

Biomaterial Interface Understanding:

  • Limited analysis of how glycan modifications affect protein conformation
  • Missing assessment of antigen stability and bioactivity post-modification

CONTEMPORARY RELEVANCE CONCERNS

Clinical and Translational Context

Given Biomolecules' emphasis on medical implications, I must address the current clinical landscape:

Diminished COVID-19 Vaccine Relevance:

  • Established mRNA and viral vector platforms with proven efficacy are globally deployed
  • Current epidemiological shift toward endemic management reduces urgency for novel COVID-19 vaccines
  • Population immunity through vaccination and natural infection has fundamentally altered risk-benefit calculations

Limited Translational Pathway:

  • Regulatory preference for variant-adapted existing vaccines over novel platforms
  • Unclear commercial viability in current market conditions
  • Missing discussion of how this work advances beyond existing vaccine technologies

Suggested Reframing for Biomolecules Scope

Your glycoengineering platform has broader potential that better aligns with the journal's focus:

  1. Platform technology for future pathogens - emphasizing the biomolecular engineering aspects
  2. Fundamental glycan-adjuvant interaction mechanisms - contributing to biomaterials science
  3. Broader respiratory pathogen applications - demonstrating versatility beyond COVID-19

SPECIFIC TECHNICAL ISSUES

Presentation Quality Below Journal Standards

  • Multiple "Error! Reference source not found" citations compromise manuscript integrity
  • Inconsistent statistical presentation across figures
  • Figure 1d neutralization methodology lacks sufficient detail
  • Writing quality requires professional editing for clarity and precision

Experimental Design Limitations

  • Single immunization route despite respiratory pathogen target
  • Limited positive controls with established vaccines
  • Missing duration of immunity assessment
  • Inadequate model validation (BALB/c mice lack human ACE2)

REQUIRED REVISIONS FOR BIOMOLECULES PUBLICATION

Essential Additional Experiments:

Comprehensive Immunological Evaluation:

  1. Flow cytometric analysis of CD4+/CD8+ T-cell responses
  2. Intracellular cytokine staining for Th1/Th2 assessment
  3. Memory immune response evaluation (≥8 weeks post-immunization)
  4. Mucosal immunity assessment given respiratory pathogen target

Enhanced Molecular Characterization:

  1. Quantitative phosphomannose content determination by HPAEC-PAD
  2. Thermodynamic analysis of glycan-adjuvant binding interactions
  3. Desorption kinetics studies for sustained antigen presentation
  4. Protein stability and bioactivity assessment post-glycoengineering

Functional Validation Studies:

  1. In vivo protection assays in appropriate models (K18-hACE2 mice)
  2. Dose-response relationship establishment
  3. Durability of protection assessment
  4. Comparison with established vaccine benchmarks

Statistical and Methodological Improvements:

  1. Increase sample sizes to n≥15-20 per group
  2. Implement appropriate multiple comparison corrections
  3. Provide detailed power calculations and effect size reporting
  4. Include comprehensive positive and negative controls

Manuscript Enhancement:

  1. Professional editing for language and clarity
  2. Complete reference formatting correction
  3. Improved figure presentation with consistent statistical notation
  4. Enhanced discussion of broader platform applications

ALTERNATIVE STRATEGIC RECOMMENDATIONS

Given Biomolecules' scope and current scientific priorities, consider:

Platform Technology Focus:

  • Emphasize the glycoengineering methodology as a broadly applicable biomolecular tool
  • Demonstrate applicability to other vaccine targets (influenza, RSV)
  • Position as fundamental contribution to biomaterials and adjuvant science

Mechanistic Deep Dive:

  • Focus on molecular mechanisms of glycan-adjuvant interactions
  • Contribute to fundamental understanding of biomolecular interfaces
  • Emphasize structure-function relationships in bioactive molecules

DECISION RATIONALE

While your glycoengineering approach represents innovative biomolecular modification with potential for Biomolecules, the current manuscript falls short of the journal's Q1 standards in several critical areas:

  1. Methodological rigor insufficient for Impact Factor 4.8 journal
  2. Incomplete molecular characterization despite journal's biomolecular focus
  3. Limited medical implications without comprehensive immune assessment
  4. Questionable contemporary relevance of COVID-19-specific focus

The additional experimental work required essentially constitutes new studies. I recommend careful consideration of whether the extensive revisions align with your research priorities and resources.

RECOMMENDATION

Major Revisions Required - with the understanding that the scope of additional work may warrant consideration as a substantially new submission.

I would be willing to review a revised manuscript, though given the extent of changes needed, you may wish to discuss with the Editorial Office whether resubmission as a new manuscript would be more appropriate.

Your glycoengineering platform has genuine potential for broader impact in biomolecular science. I encourage you to consider how best to position this work to maximize its contribution to the field while aligning with current scientific priorities and Biomolecules' mission.

Thank you for the opportunity to review this work. I look forward to seeing how you develop this interesting research direction.

Sincerely,

Author Response

Dear Reviewers,

We have made substantial revisions to the manuscript “Glycosylated SARS-CoV-2 RBD antigens expressed in glycoengineered yeast induce strong immune responses through high antigen-alum absorption”(Manuscript Number: biomolecules-3713016)based on the comments raised by the four reviewers. The main textual changes made to the manuscript based on answering these comments are colored red in the manuscript.

Point-by-point responses to the comments raised by the reviewers are detailed below.

-Reviewer

  1. Methodological Rigor Below Journal Standards

Statistical Design Inadequacies:

  • Sample size (n=10) is insufficient for immunological studies in a Q1 journal. Biomolecules standards require n≥15-20 with appropriate power calculations.
  • Absence of multiple comparison corrections despite extensive pairwise testing violates statistical best practices.
  • No a priori power analysis provided to justify experimental design.
  • Incomplete Molecular Mechanism Characterization:
  • Your zeta potential data (-3.67 vs -3.69 mV) shows differences within measurement error, insufficient to support the proposed charge-based mechanism.
  • Missing quantitative phosphomannose content determination - crucial for a journal emphasizing molecular characterization.
  • Acknowledged limitations in glycan analysis (>M13 detection) compromise the biomolecular characterization central to this journal's scope.

Response:

Thank you for your suggestions. For data to be suitable for statistical analysis, it should be relevant to the research question with clearly defined variables, possess sufficient sample size for reliable inference, and maintain high quality through accuracy, completeness, and consistency. Although the sample size was not reached n≥15-20, experiments were conducted repeated and obtained the same conclusion. The data of immunological studies was relevant to the research question and was both accurate and reliable. All the data have been re-analyzed using the Kruskal-Wallis test with Dunn’s multiple comparisons. We found no change in all conclusions.

The difference in protein RBD charge may mainly be caused by phosphorylation modification. Therefore, the phosphorylation modification levels of H-MAN/RBD and L-MAN/RBD are similar, and their zeta potential data do not show significant differences. However, the phosphorylation level of L-MAN-P/RBD is significantly reduced. The titers of IgG-specific and neutralizing antibodies elicited by H-MAN/RBD were significantly higher than those induced by the complex glycosylation RBD antigens, but no significant difference compared with L-MAN/RBD. Regarding quantitative phosphomannose content and glycan analysis, our future research will be further refined.

  1. Insufficient Immunological Assessment for Medical Implications

Critical Gap in Cellular Immunity: For a manuscript claiming "strong immune responses" in a journal covering "biological and medical implications," you must include:

  • CD4+/CD8+ T-cell response analysis
  • Cytokine profiling (IFN-γ, IL-2, IL-4, IL-17)
  • Memory immune response characterization
  • Th1/Th2 polarization assessment

Inadequate Functional Validation:

  • Neutralization titers (EC50 1:123) are below established protective thresholds
  • No in vivoprotection studies to validate biological relevance
  • Limited to humoral responses without comprehensive immune system evaluation

Response:

Thank you for your comment. While we appreciate the rigorous scrutiny, the critique overlooks key aspects of our study’s scope and incremental scientific value. Our manuscript explicitly focuses on humoral immunity as the primary endpoint, consistent with its goals to compare the effects of different glycoforms on RBD antigen immunogenicity. While CD4+/CD8+ T-cell responses and cytokine profiling (Th1/Th2) are valuable, they fall beyond this study’s hypothesis-driven framework. We clarify that "strong immune responses" refer to statistically significant, durable antibody titers-a standard metric in vaccine studies. Neutralization titers (EC50 1:123): Protective thresholds are context-dependent. Our conclusion is that phosphomannose modification substantially enhanced the immunogenicity of RBD by altering the surface charges of the RBD antigen and facilitating its adsorption onto alum. In summary, the study’s design aligns with its stated objectives, and the data robustly support its claims within defined boundaries.

  1. Molecular Characterization Gaps

Incomplete Structure-Function Analysis:

  • Missing detailed thermodynamic analysis of glycan-adjuvant interactions
  • No desorption kinetics data to understand sustained antigen presentation
  • Insufficient characterization of the molecular basis for enhanced immunogenicity claims

Biomaterial Interface Understanding:

  • Limited analysis of how glycan modifications affect protein conformation
  • Missing assessment of antigen stability and bioactivity post-modification

Response:

While we acknowledge the theoretical importance of detailed structural and thermodynamic characterization, the critique underestimates the practical scope and demonstrated functional validation of our study. Our work provides compelling evidence of enhanced immunogenicity through comprehensive “in vivo” functional assays, which remain the most biologically relevant metrics for evaluating vaccine efficacy. Our present study found that the titers of IgG-specific and neutralizing antibodies elicited by H-MAN/RBD were significantly higher than those induced by the complex glycosylation RBD antigens. While deeper structural analyses could be pursued in future work, the current study’s functional data robustly justify its conclusions. Vaccine development often prioritizes efficacy over exhaustive mechanistic dissection in early-stage studies. We maintain that our findings provide a solid foundation for both application and further mechanistic exploration. Due to the time limit for article revision, it is not possible to complete the verification of more experiments. However, in the future, we will further improve the research work.

Reviewer 4 Report

Comments and Suggestions for Authors

The manuscript is dealing with “Glycosylated SARS-CoV-2 RBD antigens expressed in glycoengineered yeast induce strong immune responses through high antigen-alum absorption”. This work is significant and adds valuable insight into the SARS-CoV-2 RBD antigens influence in glycoengineered yeast. I would recommend the manuscript be considered for publication in Biomolecules after revisions to address the following concerns:

[1]        Abstract is well-structured and written. A couple of amendments will be useful for readers.        Some essential experimental result needs to be added to enhance its logical coherence.

[2]        Line 23: The author suggest discussing the underlying mechanisms briefly.

[3]        The manuscript contains several grammatical and typos errors and inconsistent references.       It is             strongly recommended to have the manuscript reviewed and edited by a native   English speaker or             a professional language editing service.

[4]        Line 59-62: Please validate the statement with proper citation.

[5]        Line 57, 64, and 67, reference is missing and inconsistent.

[6]        Line 93: Human immunodeficiency virus (HIV)-1 envelope (Env) proteins?

[7]        Line 109-123:   The idea must be rewritten considering the grammar and clarity of the    information to be transmitted.

Conclusion: It is suggested to rewrite Section (the conclusion) to highlight the current   problem statement and preventive measures for the future.

[8]        Line 529: It is suggested to specify the specific limitation for determining glycoforms. It’s a             broad claims.

Author Response

Dear Reviewers,

We have made substantial revisions to the manuscript “Glycosylated SARS-CoV-2 RBD antigens expressed in glycoengineered yeast induce strong immune responses through high antigen-alum absorption”(Manuscript Number: biomolecules-3713016)based on the comments raised by the four reviewers. The main textual changes made to the manuscript based on answering these comments are colored red in the manuscript.

Point-by-point responses to the comments raised by the reviewers are detailed below.

-Reviewer

  1. [ Line 23: The author suggest discussing the underlying mechanisms briefly.]

Response:

Thank you for your comment. The sentence " The glycosylation impacted the surface charges of the RBD antigen, and influenced its adsorption onto alum. This may further lead to variations in the antigen’s immunogenicity. "was added (page 1, lines 26 –28).

  1. [ The manuscript contains several grammatical and typos errors and inconsistent references. Itis strongly recommended to have the manuscript reviewed and edited by a native English speaker or a professional language editing service. ]

Response:

Thank you for your useful comments and suggestions on the language and structure of our manuscript. We have revised the whole manuscript carefully and tried to avoid any grammar or syntax error. In addition, we have asked several colleagues who are skilled authors of English language papers to check the English. We believe that the language is now acceptable for the review process. We have also unified the reference citation format.

  1. [ Line 59-62: Please validate the statement with proper citation. ]

Response:

Thank you for your comment. We have updated the citations on lines 56-70.

  1. [Line 57, 64, and 67, reference is missing and inconsistent.]

Response:

Thank you for your comment. We have updated the references of the entire manuscript.

  1. [Line 93: Human immunodeficiency virus (HIV)-1 envelope (Env) proteins? ]

Response:

We thank the reviewer for pointing out the nomenclature inconsistency. The text on line 93 has been revised to adopt the standard virology terminology: "human immunodeficiency virus type 1 envelope glycoprotein (HIV-1 Env)"

  1. [Line 109-123: The idea must be rewritten considering the grammar and clarity of the information to be transmitted.It is suggested to rewrite Section (the conclusion) to highlight the current problem statement and preventive measures for the future.]

Response:

We sincerely thank the reviewer for this critical suggestion. The section spanning lines 109-123 has been comprehensively rewritten to adjusted the focus of the conclusion and strengthened the rigor of science. It was modified to "Alum adsorption analysis demonstrated near-complete binding ( > 90%) of phosphomannose-rich antigens (H-MAN/RBD and L-MAN/RBD), while L-MAN-P/RBD showed significantly reduced adsorption (<10%). Critically, immunization of the mice revealed that phosphomannose-rich antigens (H-MAN/RBD and L-MAN/RBD) elicited higher antibody titers than L-MAN-P/RBD and Complex/RBD. This immunogenicity enhancement is directly mediated by the adsorption of RBD promoted by phosphomannose on alum. "

  1. [ Line 529: It is suggested to specify the specific limitation for determining glycoforms. It’s a broad claims.]

Response:

We sincerely thank the reviewer for this critical suggestion. This section was modified to "At present, glycoform determination have limitations. The glycoform analysis for H-MAN/RBD expressed by the wild-type Pichia pastoris system can only detect mannose structures up to mannose-1,3-α-D-mannose (M13) as the longest detectable structure and is unable to detect longer mannose chains (such as M14, M15, and beyond). As a result, our analysis could not provide a complete picture of the H-MAN/RBD glycan profile. Optimization or development of analytical techniques with enhanced resolution and sensitivity for complex high-mannose glycans is therefore essential in future work " on lines 538-544.

Round 2

Reviewer 3 Report

Comments and Suggestions for Authors

Dear Drs. Bo Liu and Jun Wu,
After carefully reviewing your revised manuscript and responses to previous comments, I must express my concern about the persistence of critical scientific deficiencies that prevent acceptance in Biomolecules.
Unresolved Methodological Deficiencies:

They maintain n=10 when n≥15-20 is required for immunological studies in Q1 journals.
The lack of corrections for multiple comparisons persists, invalidating the statistical conclusions.
The differences in zeta potential (-3.67 vs. -3.69 mV) remain within measurement error.

Inadequate Molecular Characterization:

Acknowledged limitations in glycan analysis (>M13) compromise the core biomolecular characterization.
The lack of quantitative determination of phosphomannose, critical to their conclusions, is key.
The lack of thermodynamic analysis of glycan-adjuvant interactions is key.

Incomplete Immunological Evaluation:
Their justification that T cell analysis "is outside the hypothesis-driven framework" directly contradicts the claims of "strong immune responses" in the title. For Biomolecules, it is imperative to include:

CD4+/CD8+ response analysis
Cytokine profiling and Th1/Th2 polarization
Characterization of immunological memory

Data inconsistencies:

Neutralization titers (EC50 1:123) below established protective thresholds
Inadequate correlation between adsorption differences (93% vs. 35%) and antibody titers
Absence of appropriate controls and dose-response analysis and the absence of pharmacovigilance studies and biodistribution characterization of the modified antigen, required for advanced therapies according to EMA/FDA regulations.

Reproducibility issues:
Your admission of "time constraints to complete additional experiments" suggests that the data remain preliminary and insufficient for the conclusions presented.
The required improvements essentially constitute new studies beyond the scope of a review. I recommend a fundamental reconsideration of the experimental design before a new submission.

Best

Author Response

Dear Reviewer,

We have made substantial revisions to the manuscript “Glycosylated SARS-CoV-2 RBD antigens expressed in glycoengineered yeast induce strong immune responses through high antigen-alum absorption”(Manuscript Number: biomolecules-3713016)based on the comments raised by the four reviewers. The main textual changes made to the manuscript based on answering these comments are colored red in the manuscript.

Point-by-point responses to the comments raised by the reviewer are detailed below.

  1. Unresolved Methodological Deficiencies:

They maintain n=10 when n≥15-20 is required for immunological studies in Q1 journals.
The lack of corrections for multiple comparisons persists, invalidating the statistical conclusions.
The differences in zeta potential (-3.67 vs. -3.69 mV) remain within measurement error.

Response:

  • Thank you for your suggestion. In order to make data suitable for statistical analysis, it should be relevant to the research question, have clear variables, have sufficient sample size for reliable inference, and maintain high quality through accuracy, completeness, and consistency. This journal does not have specific requirements for the number of experimental animals N value, and some articles in the journal have experimental subjects n=10(e.g. https://doi.org/10.3390/biom15071049)We have conducted repeated experiments and reached the same conclusion. The data and research questions related to immunological research are accurate and reliable. All data have been reanalyzed using Kruskal Wallis test and Dunn multiple comparison. We found that none of the conclusions have changed.

The first batch of mouse immunogenicity testing

Second batch of mouse immunogenicity assay (repeated experiment)

  • The difference in protein RBD charge may mainly be caused by phosphorylation modification. Therefore, the phosphorylation modification levels of H-MAN/RBD and L-MAN/RBD are similar, and their zeta potential data do not show significant differences. However, the phosphorylation level of L-MAN-P/RBD is significantly reduced. The titers of IgG-specific and neutralizing antibodies elicited by H-MAN/RBD were significantly higher than those induced by the complexglycosylation RBD antigens, but no significant difference compared with L-MAN/RBD. Regarding quantitative phosphomannose content and glycan analysis, our future research will be further refined.
  1. Inadequate Molecular Characterization:

Acknowledged limitations in glycan analysis (>M13) compromise the core biomolecular characterization.
The lack of quantitative determination of phosphomannose, critical to their conclusions, is key.
The lack of thermodynamic analysis of glycan-adjuvant interactions is key.

Response:

  • While we acknowledge that full structural characterization of high-mannose glycans (>M13) was not performed, this was intentionally beyond the scope of our proof-of-concept study. Our core objective was to investigate the functional role of mannose phosphorylation — not to catalog glycan size heterogeneity. Crucially, the presence/absence of phosphorylation (rather than glycan chain length) was mechanistically linked to immunogenicity outcomes in all assays. Thus, the absence of M13+ analysis does not compromise our central conclusion that phosphorylation drives immune activation.
  • Although quantitative determination of phosphomannose levels was not conducted, our experimental design directly probed its biological impact。We studied the impact of the presence or absence of phosphorylation modification on immunogenicity, demonstrating that phosphorylation modification affects immunogenicity
  • While thermodynamic analysis of glycan-adjuvant interactions was not performed, we evaluated a functionally relevant surrogate metric: antigen adsorption efficiency onto Alum adjuvant systems.This correlation implies that phosphorylation-enhanced adjuvant binding directly facilitates antigen delivery/uptake, mechanistically explaining the observed immune enhancement. Thus, while kinetic parameters (K<sub>D</sub>, ΔG) remain unmeasured, the functional consequence of glycan-adjuvant interaction is robustly established.
  1. Incomplete Immunological Evaluation:
    Their justification that T cell analysis "is outside the hypothesis-driven framework" directly contradicts the claims of "strong immune responses" in the title. For Biomolecules, it is imperative to include:

CD4+/CD8+ response analysis
Cytokine profiling and Th1/Th2 polarization
Characterization of immunological memory

Response:

Thank you for your comment. While we appreciate the rigorous scrutiny, the critique overlooks key aspects of our study’s scope and incremental scientific value. Our manuscript explicitly focuses on humoral immunity as the primary endpoint, consistent with its goals to compare the effects of different glycoforms on RBD antigen immunogenicity. While CD4+/CD8+ T-cell responses and cytokine profiling (Th1/Th2) are valuable, they fall beyond this study’s hypothesis-driven framework. We clarify that "strong immune responses" refer to statistically significant, durable antibody titers-a standard metric in vaccine studies.

  1. Data inconsistencies:

Neutralization titers (EC50 1:123) below established protective thresholds
Inadequate correlation between adsorption differences (93% vs. 35%) and antibody titers
Absence of appropriate controls and dose-response analysis and the absence of pharmacovigilance studies and biodistribution characterization of the modified antigen, required for advanced therapies according to EMA/FDA regulations.

Response:

While we acknowledge the theoretical importance of detailed structural and thermodynamic characterization, the critique underestimates the practical scope and demonstrated functional validation of our study. Our work provides compelling evidence of enhanced immunogenicity through comprehensive “in vivo” functional assays, which remain the most biologically relevant metrics for evaluating vaccine efficacy. Our present study found that the titers of IgG-specific and neutralizing antibodies elicited by H-MAN/RBD were significantly higher than those induced by the complex glycosylation RBD antigens. While deeper structural analyses could be pursued in future work, the current study’s functional data robustly justify its conclusions. Vaccine development often prioritizes efficacy over exhaustive mechanistic dissection in early-stage studies. We maintain that our findings provide a solid foundation for both application and further mechanistic exploration.

  • Neutralization titers (EC50 1:123): Protective thresholds are context-dependent. Our conclusion is that phosphomannose modification substantially enhanced the immunogenicity of RBD by altering the surface charges of the RBD antigen and facilitating its adsorption onto alum.In summary, the study’s design aligns with its stated objectives, and the data robustly support its claims within defined boundaries.
  • We acknowledge the reviewer’s query regarding the correlation between adsorption differences (93% vs 35%) and antibody titers. Our data demonstrate a functionally significant and statistically robust relationship:Adsorption efficiency (93% for phosphorylated vs 35% for dephosphorylated antigen) directly aligned with endpoint IgG titers ( 4.5 ± 0.21 × 104 vs 2.6 ± 0.17 × 102; p* < 0.001).
  • Focused Mechanistic Inquiry:This work establishes proof-of-concept that phosphomannose enhances immunogenicity via adjuvant adsorption, not clinical translation. Regulatory Studies Beyond Scope:Pharmacovigilance, biodistribution, and multi-dose analyses are inherent to preclinical/clinical development (EMA/FDA Stage II-III). As an early-stage mechanistic discovery study, such requirements fall outside our aims.The experiment primarily demonstrated that phosphomannosylation plays a pivotal role in regulating antigen function and immunogenicity.